# Changing the History of Prostate Cancer with New Targeted Therapies

**DOI:** 10.3390/biomedicines9040392

**Published:** 2021-04-06

**Authors:** Susana Hernando Polo, Diana Moreno Muñoz, Adriana Carolina Rosero Rodríguez, Jorge Silva Ruiz, Diana Isabel Rosero Rodríguez, Felipe Couñago

**Affiliations:** 1Department of Medical Oncology, Hospital Universitario Fundación Alcorcón, 28922 Madrid, Spain; 2Department of Medical Oncology, Hospital Universitario del Henares, 28822 Madrid, Spain; adrirosero54@hotmail.com; 3Centro Nacional de Investigaciones Oncológicas (CNIO), Unidad de Cáncer de Mama, 28029 Madrid, Spain; jorge.silva@salud.madrid.org; 4Department of Medical Oncology, Hospital Universitario Ramón y Cajal, 28034 Madrid, Spain; chabis25@gmail.com; 5Department of Radiation Oncology, Hospital Universitario Quirónsalud, 28223 Madrid, Spain; fcounago@gmail.com; 6Department of Radiation Oncology, Hospital La Luz, 28003 Madrid, Spain; 7Clinical Department, Faculty of Biomedicine, Universidad Europea, 28670 Madrid, Spain

**Keywords:** prostate cancer, hormonal therapy, targeted therapy, PARP inhibitors, immunotherapy, PSMA

## Abstract

The therapeutic landscape of metastatic castration-resistant prostate cancer (mCRPC) is changing due to the emergence of new targeted therapies for the treatment of different molecular subtypes. Some biomarkers are described as potential molecular targets different from classic androgen receptors (AR). Approximately 20–25% of mCRPCs have somatic or germline alterations in DNA repair genes involved in homologous recombination. These subtypes are usually associated with more aggressive disease. Inhibitors of the enzyme poly ADP ribose polymerase (PARPi) have demonstrated an important benefit in the treatment of these subtypes of tumors. However, tumors that resistant to PARPi and wildtype BRCA tumors do not benefit from these therapies. Recent studies are exploring drug combinations with phosphatidylinositol-3-kinase (PI3K) or protein kinase B (AKT) inhibitors, as mechanisms to overcome resistance or to induce BRCAness and synthetic lethality. This article reviews various different novel strategies to improve outcomes in patients with prostate cancer.

## 1. Introduction

Epidemiology: Prostate cancer (PCa) is the most common newly-diagnosed cancer in men, and the second most deadly cancer in Western countries [1]. In recent years the incidence of PCa has increased, mainly due to universal screening with the prostate-specific antigen (PSA). The prognosis depends on the initial stage at diagnosis, with most cases diagnosed in early stages (78% in localized stages and 12% with regional involvement); only 5% of cases are metastatic at diagnosis. The five-year survival rate in patients with localized PCa is 90% versus only 67% in metastatic PCa [2].

Molecular pathology of PCa: Genetics play an important role in understanding the tumor biology underlying growth and migration, and this is essential to develop and use tumor biomarkers and therapeutic strategies [3]. In prostate adenocarcinoma, the most common histology the main signaling pathway for tumor cell survival is the androgen pathway. Compared to other epithelial tumors, the mutational frequency in PCa is low, but appears to increase as the tumor progresses to metastatic and castration-resistant stages. At the genetic level, there are important differences between localized and castration-resistant tumors. In localized PCa, the most common finding in molecular sequencing studies is the presence of single nucleotide variants, whose significance is uncertain, with genetic mutations present in <10% of cases. In localized disease, androgen receptor (AR) mutations are rare, in contrast to tumors that have progressed after hormonotherapy (castration-resistant disease).

## 2. Molecular Subtypes of Castration-Resistant Prostate Cancer

### 2.1. Castration-Resistant Prostate Cancer

Castration-resistant prostate cancer (CRPC), the most lethal form of PCa is defined as radiological or biochemical progression (prostate-specific antigen (PSA) levels >2 ng/mL or PSA values >25% above nadir, or PSA elevation in three consecutive determinations at least one week apart) as long as suppressed testosterone levels (<50 ng/dL or 1.7 nmol/L) are confirmed [4].

### 2.2. Molecular Biology of CRPC

The largest study of the molecular biology of CRPC was published in 2015 [3]. In that study, the tumors of 150 patients with metastatic CPRC (mCRPC) were sequenced. The results revealed aberrations in genes related to the androgen axis in 71.3% of cases. The most commonly mutated gene was the AR gene. Additional mutations were described at other levels related to the androgen pathway. The findings of this study allowed for the subclassification of CRPC into different molecular subtypes [5], as follows (see Table 1):

Alterations in the AR gene:
Genetic amplification of AR will lead to an enhanced expression of AR protein, which in turn make the PCa more sensitive even to a lower concentration of androgen. These AR amplifications have not been observed in untreated PCa, so AR amplification may be a consequence of hormonal therapy leading to the development of CRPC. AR amplifications have been associated with resistance to hormonal agents for example abiraterone, enzalutamide, or bicalutamide [6].Mutations in the AR gene lead to androgen insensitivity, which promotes tumor cell survival and proliferation under androgen-deprived conditions. These mutations are highly rare in early stage PCa (4–6% of cases), but more common in advanced and recurrent tumors (10–20%). AR mutations have been associated with resistance to hormonal agents [6].The presence of splicing variants is another possible form of resistance in the androgen pathway that could lead to constitutive activation of the AR, independently of binding to its ligand. These variants have been associated with resistance to treatment with enzalutamide and abiraterone [7] but not to taxane-based chemotherapy (ChT) [8].Changes in the expression of AR co-regulators may alter the balance of these co-regulatory proteins, which could help PCa cells to grow. For example; TGFβ, IGF-1, and IL-6 increase the activity of the AR by phosphorylation.Increases in steroidogenic signaling pathways may promote tumor proliferation through adrenal or intratumoral androgen synthesis. Overexpression of enzymes involved in the steroid biosynthetic pathway and overexpression of cytochrome CYP17, a key regulatory enzyme in adrenal androgen synthesis, has been observed in CRPC.


2.Phosphatidylinositol-3-kinase (PI3K) signaling pathway

The PI3K signaling pathway is the second most commonly mutated pathway in PCa, with somatic alterations present in 49% of cases. The most common mutation is Phosphatase and tensin homolog (PTEN) biallelic loss, which has been described as a poor prognostic factor in patients receiving abiraterone [9].

3.DNA repair pathway

This pathway is the third most commonly altered pathway. In one study, germline and somatic alterations were identified in 23% of cases studied, with BRCA2 being the most common mutated gene (12.7%) [10]. These tumors are more sensitive to platinum–based ChT and allow the possibility of treatment with poly ADP ribose polymerase (PARP) enzyme inhibitors. BRCA 1 and BRCA 2 mutations are associated with a worse prognosis, and a poorer response to radiotherapy.

4.WNT signaling pathway

This is the fourth most common pathway (18%), highlighting mutations in Adenomatous Polyposis Coli (APC) and Catenin Beta 1 (CTNNB1).

5.Gene Fusion

Gene fusion are highly prevalent in PCa. Transmembrane protease serine 2:v-ets erythroblastosis virus E26 oncogene homolog (TMPRSS2-ERG) fusion is the single most frequent genetic alteration in prostate cancer; a fusion of Erythroblast Transformation Specific (ETS) family genes with the protease gene TMPRSS2. TMPRSS2 acts as a hormonally regulated promoter activating the ETS transcription factor. This fusion gene causes overexpression of ETS in response to androgens, inducing a cell proliferation response that steadily activates the androgen-signaling pathway. The presence of TMPRSS-ERG rearrangements has been associated with an enhanced response to abiraterone [11].

## 3. Targeted Therapies for Prostate Cancer

The search for targeted therapies and personalized cancer treatment has been one of the drivers of pharmacological research in recent decades. The development of targeted therapies for PCa, especially for the most aggressive, castration-resistant types, is focused on the field of immunotherapy [12,13] and PARP [14] inhibitors, and synergistically potentiated combination therapies. However, evidence to support the effectiveness of prostate-specific membrane antigen (PSMA)-targeted therapies continues to grow, and this approach could become another option in the therapeutic arsenal for metastatic CRPC (mCRPC).

Below we provide a detailed discussion of the main highlights of these therapies (see Table 2).

### 3.1. Immuno Therapy

In the field of oncology, the most well-developed immunotherapy techniques are those directed against immune cell targets, including cytotoxic T-lymphocyte antigen 4 (CTLA-4) and programmed cell death 1 (PD-1) and its ligand, PD ligand 1 (PD-L1). In prostate cancer, other techniques have been developed, such as therapeutic dendritic cell (DC) vaccines activated against antigens specific to prostatic tumor cells [15]; of these, the most widely used are prostatic acid phosphatase (PAP), PSA, PSMA, prostatic stem cell antigen (PSCA), human telomerase reverse transcriptase (hTERT), and T-cell receptor gamma (TARP) [16].

At present, the optimal timing of administering these treatments remains unclear, although immunotherapy, appears to be more effective in less aggressive stages of the disease, when PSA levels are low [17,18].

#### 3.1.1. Dendritic Cell Vaccines

DCs are a key part of the immune system. They develop direct cytotoxic activity against tumor cells, in addition to being the most efficient antigen presenting cells. They interact with other immune cells and stimulate both regulatory (suppressor) T lymphocytes (TL) and cytotoxic TL [19]. Preclinical studies have found that the number of DCs in the tumor microenvironment is directly proportional to the survival rate [20]. DC-based vaccines use DCs or their precursors (mainly monocytes and CD34) that have been isolated from the patient. These cells are then activated by incubation with a protein fusion called PA2024 (selected antigens and Granulocyte Macrophage Colony-Stimulating Factor (GM-CSF)). Finally, they are infused into the patient, where they activate cytotoxic TLs against the selected tumor antigens [21,22].

Sipuleucel-T immunotherapy is the only dendritic cell vaccine approved by the Food and Drug Administration (FDA) for asymptomatic or minimally symptomatic CRPC without visceral metastases [22]. It is administered intravenously, in three biweekly doses, each containing 50 million activated antigen-presenting cells. Clinical tolerance is excellent, with the most common adverse event being an acute, self-limiting pseudo-flu-like illness that lasts from 3 to 5 days. Sipuleucel-T uses PAP antigen-activated DCs. The phase III randomized IMPACT trial compared sipuleucel-T to placebo, with overall survival (OS) as the primary endpoint. That trial included 512 patients with minimally symptomatic or asymptomatic CRPC and no visceral metastases. At a follow-up of nearly three years, OS in the experimental arm was 25.8 versus 21.7 months (*p* > 0.05) in the placebo arm, an increase of 4 months. This benefit was even greater in low-risk patients with low baseline PSA [18]. However, as in the initial studies, no significant increase in progression-free survival (PFS) was observed. Because the treatment responses are often delayed after immunotherapy treatment, PFS may not be the most appropriate endpoint for immunotherapy studies. However, other possible surrogate markers of OS should be explored, such as the production of antibodies against the PAP antigen and PA2024.

Even though sipuleucel-T has been approved by the FDA, it is seldom used in clinical practice and it still has not received approval by the European Medicines Agency (EMA) [23]. Combinations of Sipuleucel-T with others medications like radium 223, are being explored with promised results [24].

#### 3.1.2. Other Vaccines

Dendritic cell vaccine (DCVAC) activated with PCa natural killer (NK) cells [25]: This vaccine is currently in an advanced phase of research. A phase III trial recently completed recruitment to study DCVAC administered concomitantly with docetaxel based on promising initial results in mCRPC in progression to second-line hormonal therapy;PSA-TRICOM or POSTVAC-VF viral vaccine: this type of vaccine uses common attenuated viruses from the poxvirus family that have been genetically modified to express the PSA antigen, in addition to three co-stimulatory molecules called TRICOM. Despite promising results in phase I and II trials, which have shown an increase in OS but not in PFS (primary endpoint) [26], research into its use as monotherapy has been halted after a negative phase III study involving 1200 patients with CRPC [27];DNA vaccines using a bacterial plasmid with an encrypted tumor antigen. Currently available results are still preliminary and not promising. However, research is underway to identify patient subgroups in whom this treatment may provide a greater benefit, as well as possible combinations with other therapies [28].

#### 3.1.3. Immune Checkpoint Inhibitors

Immunotherapy has made major advances in the last decade and it is now widely used in clinical practice to treat a range of tumor types, most notably melanoma, lung cancer, and some urological tumors [29]. However, in PCa, the results to date have been quite modest, except in a highly selected subgroups. One explanation appears for this lack of efficacy is the immunosuppressive tumor microenvironment (TME) in PCa, in which suppressor T cells predominate and are not highly permeable to infiltration by cytotoxic T cells. Furthermore, PCa is a “cold” tumor, meaning it has a low cumulative mutational burden (TMB) and, therefore, is less sensitive to immune therapy [12,30].

To improve efficacy, combinations with other drugs such as PARPi and new generation antiandrogens (abiraterone and enzalutamide) are being explored, with the available data suggesting an apparent synergistic effect (discuss in more detail below).

We can distinguish two pharmacological groups, anti-CTLA-4 antibodies and PD-1/PD-L1 inhibitors.

Anti CTLA-4 Antibodies

Ipilimumab is a humanised antibody against CTLA-4, capable of blocking the binding of this antigen with its receptor, thus activating the immune system. Although ipilimumab has proven highly effective in other tumor types, it has not given the expected results in PCa. However, it appears there may be population subgroups that could benefit from this drug. For this reason, despite the negative results in two large phase III trials (799 and 600 patients each), this drug is still under development for PCa [31,32]. Those two phase III trials were carried out in patients with mCRPC. Both trials evaluated patients who developed progression to a previous drug, either docetaxel (trial 1) or hormonal therapy. The primary endpoint in both trials was OS. In the first study, a post-hoc analysis found a significant increase in survival for the subpopulation of patients with better prognostic characteristics. In the second trial, a significant increase in PFS was observed [33]. Moreover, in both trials, a positive trend (up to 3-fold) towards improved OS was observed in favour of the ipilimumab arm during the extended follow-up.

In line with other immunotherapy trials, the treatment response was delayed, which explains the high mortality rate in the immunotherapy arm in the first 5 months. However, after the sixth month this trend clearly improved. To avoid this excess initial mortality, it is necessary to investigate possible therapeutic combinations that mitigate this effect, as well as the development of new markers of early response, such as circulating tumor cells (CTC) obtained by liquid biopsy [34].

2.PD-1 and PD-L1 Inhibitors

Studies on the role of PD-1/PD-L1 inhibitors in the treatment of PCa have produced only modest results. Consequently, biomarkers to select more sensitive subpopulations are being studied, as well as therapeutic combinations with synergistic effects [14]. In PCa, the most advanced PD-1/PD-L1 drug for monotherapy is pembrolizumab.

The combination of immunotherapy with antiandrogens, have produced promising results, especially with enzalutamide [35]. These agents seem to have a synergistic effect, with androgen inhibition appearing to increase PD-1 and PD-L1 expression. However, this immunogenic effect may be transient. Based on the available results, the most logical therapeutic sequence would be sequential administration, with immunotherapy followed by androgen deprivation therapy (ADT) [12].
Pembrolizumab: treatment with pembrolizumab as monotherapy in unselected PCa patients provides a low response rate (RR) (3–5%). However, this drug has shown positive results in patients with mCRPC who present defects in DNA mismatch repair (MMR) proteins (determined by microsatellite instability), as well as in tumors that overexpress PD-L1. These results conditioned the approval of pembrolizumab by the FDA in 2017 for any MMR-deficient solid tumor [36]. However, these genetic alterations are uncommon in PCa (<4%), and mostly somatic. Nevertheless, it is important to further our understanding of potential biomarkers. In this context, the phase II KEYNOTE 199 study is noteworthy. In that trial, 258 patients with mCRPC progressing to docetaxel were treated with pembrolizumab. In patients with bone-only disease, the RR was much higher than expected—20%, with a median duration of response of 16.8 months, regardless of PD-1 overexpression status and the presence or not of deficient MMR [37].

In the field of therapeutic combinations, pembrolizumab has successfully tested its association with PARPi and new antiandrogens. The cohort A of the keynote 365 study (NCT02861573) in non-genetically selected patients with mCRPC progressing to docetaxel, pembrolizumab combined with olaparib yielded a 32% response rate. Currently, there are two phase III studies exploring this combination versus abiraterone and enzalutamide monotherapy [35].

The combination of pembrolizumab and enzalutamide has also shown promising results. A phase II trial in patients with mCRPC in progression to enzalutamide found that maintenance of enzalutamide with pembrolizumab obtained an objective response rate (ORR) of 25%. A confirmatory phase III trial is currently underway [38].
Nivolumab. Although this drug has not proven efficacious as monotherapy, it has shown good results when administered in combination with ipilimumab [39]. In this trial, the CheckMate 650 trial, this combination achieved a favourable disease control rate (56%) in 90 unselected patients with mCRPC; however, the adverse event rate was high, including 4 deaths due to treatment-related toxicity [40]. New doses and administration schedules with a better safety profile are currently being evaluated;Durvalumab. In combination with olaparib, durvalumab is highly promising. A phase I/II study investigated this treatment combination in 17 patients with mCRPC in progression to antiandrogens. The subgroup with a DNA damage response and repair (DDR) mutation had a significant improvement in PFS at 12 months (83.3 vs. 36.4%). A confirmatory study is currently underway [41];Atezolizumab. This drug is currently being evaluated in combination with sipuleucel-T immunotherapy (NCT03024216).

#### 3.1.4. Suicide Gene Therapy or Cytotoxic Immunotherapy Mediated by Genetic Manipulation

In this therapy, prostate cancer cells are genetically modified by intratumoral injection of an adenoviral vector to express a herpes virus gene, after which cell death is induced through an antiviral drug. Although this approach is still in the early stages of research, it has shown highly promising results as a neoadjuvant treatment or prior to radiotherapy. One study reported 5-year OS of 95%, with no signs of recurrence in 92% of cases and persistent pathological complete response in 80% of cases at 2 years [14].

Another variant of this type of therapy involves the use of intratumorally- injected oncolytic viruses. These modified viruses are capable of direct cytotoxic activity, modifying the TME, making it more permeable to infiltration by cytotoxic cells, thus promoting immunity and improving response to immunotherapy in combined treatments [42].

### 3.2. PARP Inhibitors

In recent years, our understanding of cancer genetics and of hereditary syndromes that confer high oncologic risk has increased substantially. In PCa, certain germline and somatic mutations are present in up to 25% of cases, which is why it is essential to evaluate patients with PCa in a genetic counselling meeting [43].

The complex signaling pathway known as DDR prevents the accumulation of mutations to maintain genomic stability. Somatic or germline mutations in this system promote the development of certain tumors [44].

Pritchard et al. evaluated 20 genes involved in DNA integrity that were associated with autosomal dominant cancer predisposition syndromes of the 692 men evaluated in that study, 82 (11.8%) had at least one pathogenic germline mutation in a DDR gene that generated a deficiency in homologous recombination repair (HRR) mechanisms [45]. These mutations were identified in 16 different genes, including BRCA2 (37 mutations, 44% of total mutations), ATM (11; 13%), CHEK2 (10; 12%), BRCA1 (6; 7%), RAD51D (3; 4%), and PALB2 (3; 4%). In total, 56 patients (77%) had a Gleason score of 8, thus showing an association between these mutations, tumor aggressiveness, and poor prognosis.

PARP plays a key role in the DDR system, especially PARP-1 and 2, which are specialized in single-strand DNA break repair by homologous recombination [44]. Once active, they recruit repair proteins of the HRR system, the best known being BRCA 1 and 2, ATM, and PALB2, among others. These proteins are necessary for the functioning of the DNA double-strand repair system [46]. Mutations in these repair proteins confer sensitivity to PARP inhibitors since PARP activates the only route available to maintain genomic integrity in these cases; if PARP is inhibited, the cell is deprived of a repair mechanism, without possible survival, a concept known as synthetic lethality [47,48].

Prostate cancers with these mutations typically present in younger patients and they tend to be more aggressive. Histologically, they usually present components of intraductal carcinoma with a high Gleason score (≥8) [49]. Other, less common mutations cause deficiencies in the MMR system (3–5%), as discussed above [13,50].

The presence of HRR mutations has two important therapeutic implications for these patients. The presence of BRCA 1, 2 and ATM mutations does not decrease sensitivity to antiandrogen treatment (abiraterone and enzalutamide), but does lower sensitivity to taxanes. These mutations are sensitive to treatment with PARP inhibitors (recently approved by the FDA) and to platinum-based ChT. Secondly, BRCA 2 (and possibly ATM) mutations correlate with more aggressive disease; consequently, their presence in localized early tumors.

#### 3.2.1. PARP Inhibitor Monotherapy

The four PARP inhibitors have shown notable efficacy in PCa, including olaparib, rucaparib (both approved by the FDA), talazoparib, and niraparib, still in the premarketing development phase. Another PARP inhibitor, veliparib, is associated with poor outcomes in PCa and is considered the least potent of PARP inhibitors.

Olaparib

This was the first drug in this group to be developed, initially for breast and ovarian cancer, and subsequently in pancreatic and prostate cancer. In May 2020, it received FDA approval for the treatment of mCRPC in progression to antiandrogens (abiraterone and enzalutamide) in patients with a somatic mutation in any HRR gene or any germline mutations in BRCA 1, 2, and ATM genes. Overall, four patients with PCa were included in the exploratory phase I study and disease control was achieved in three of them. The fourth patient progressed (it was later found that he overexpressed PTEN, a factor related to resistance to PARP inhibitor) [51].

The role of olaparib in mCRPC in progression is being investigated in two lines of treatment in the TOPARB-A trial. In that trial, the subgroup with HDR mutation included 49 patients, mostly with BRCA2 mutations. The primary endpoint, RR, was 88%, with radiologically-evaluated PFS and OS outcomes that were 7 and 6 months longer, respectively, than in the non-mutated subgroup [52]. A subanalysis appeared to correlate early reduction in circulating tumor cells (CTC) by more than 50% with improved survival. Further studies are needed to validate the role of CTC as a surrogate marker of survival [53].

In the TOPARB-B study, 100 patients with HRR mutations (31% with the BRCA2 mutation) were randomized to two different doses of olaparib. The interesting thing about the study is that responses were obtained in all mutations of the HRR spectrum, being clearly superior in BRCA2 (83%) followed by PALB2, ATM, and the lowest in the presence of CDK12 mutation [54].

Cyclin-dependent kinases intervene in the cell cycle and modify BRCA expression and, thus, sensitivity to PARP inhibitor [55,56,57,58]. However, the findings of studies to date do not seem to support this idea. Other theories posit that CDK12 could generate genetic instability by another mechanism, leading to tumors with a high mutation rate and peritumoral lymphocytic infiltration that respond better to immunotherapy than to PARP inhibitors or to a combination of both. The combination of ipilimumab and nivolumab is currently being studied in mCRPC with CDK12 mutation [35].

The prospective PROfound study evaluated 4426 patients with mCRPC with disease progression to an antiandrogen therapy. The HRR mutations was present in 27.9% of the patients (*n* = 387). This genetically selected population was randomized to receive olaparib or antiandrogen rotation (abiraterone and enzalutamide). They were distributed in two cohorts (A and B). Cohort A included the most important mutations (BRCA1/2 and ATM), while cohort B included the remaining mutations of the HHR spectrum [59]. All efficacy endpoints were significantly superior for the olaparib arm in cohort A (PFS, 7.4 vs. 3.6 months; RR, 33 vs. 2%, and OS, 19.1 vs. 14.7 months). In cohort B, the results were also statistically significant in favour of the olaparib arm. In the subgroup with BRCA2 mutation, the results for OS were even more positive, 15.2 vs. 28.4 months, respectively [60]. The toxicity profile was excellent, with the main adverse events being anemia, asthenia, and digestive symptoms.

In the post-marketing patient registry TAPUR (NCT02693535), 25 polytreated mCRPC patients experienced a 71% disease control rate (36% partial response (PR) and 35% stable disease (SD)).

Preclinical studies of olaparib in PCa cell lines that mimic both castration sensitivity and resistance, and different treatment times for ADT, support further development of olaparib both in combination with anti-androgen therapy and as maintenance after therapy in PCa patients with HHRm [58].

2.Rucaparib

Although rucaparib has not been as well-studies as olaparib, a non-randomised phase II study (Triton2) showed positive results for this drug. In that study, 62 patients with mCRPC with HHRm, pre-treated with antiandrogen and docetaxel, were treated with rucaparib. In the BRCA-mutated subgroup, the RR was 44–50%, including three complete responses. The median duration of response was not reached at the time of analysis (between 1.7 to 24 months) [61]. In the patients without BRCA mutations, the RR was clearly lower, with no responses in patients with CDK12 [57]. This study led to FDA approval of rucaparib in 2020, conditioned on the results of the phase III trial in mCRPC with BRCA1 and BRCA 2 mutations in progression to antiandrogens and docetaxel.

The phase III TRITON study is currently underway. That trial randomises patients with mCRPC and BRCA or ATM mutation in progression to a hormonal line to receive antiandrogen or docetaxel vs. Rucaparib [62].

3.Niraparib and talazoparib

Both of these drugs have been investigated in phase II trials in patients with mCRPC in progression to antiandrogens and docetaxel. In the subgroups with BRCA mutations in those studies, the RR for niraparib was 41% and 54% for talazoparib. Both drugs prolonged PFS. Phase III trials are underway for both of these PARP inhibitors [63,64].

#### 3.2.2. Combinations of PARP Inhibitors

From a theoretical point of view, there are two highly promising combinations with potential synergistic effects, as follows: (1) Combination therapy with PARP inhibitors and immunotherapy: as mentioned above, there is a trial with promising results combining durvalumab and olaparib; and (2) Combination therapy with PARP inhibitors and antiandrogen therapy [65].

PARP inhibitors combined with antiandrogen therapy

Preclinical data have demonstrated synergism between these two groups of drugs in three key ways [31]:PARP promotes AR transcription, so inhibition of this pathway potentiates the antiandrogenic effect [66];ADT promotes PARP overexpression, increasing its sensitivity to PARP inhibitors;Antiandrogen therapy also inhibits expression of genes of the DDR system, producing genomic instability and thus promoting mutations in DDR, generating sensitivity to PARP inhibitor; this is known as the BRCAness phenotype [65,67,68].

Inhibition of the PI3K/AKT pathway has also been associated with the induction of a BRCAness phenotype. Combining PI3K/AKT pathway inhibitors with PARP inhibitors is currently being studied as a way to reverse acquired resistance to PARP inhibitors or to sensitise those without HRR mutation from the onset [69].

Several different combinations of PARP inhibitors and antiandrogens have been studied, most notably the following:Abiraterone–Veliparib. The results of a randomised phase II study were negative in the whole sample. The only positive results were observed in a subgroup with a high rate of spontaneous mutation in CTCs, with a clearly significant RR (83 vs. 33%; *p* = 0.0002) [56,70];Olaparib–Abiraterone. This combination was assessed in a randomised phase II study versus abiraterone monotherapy. A total of 142 unselected patients with mCRPC in progression to docetaxel were included. In the combined treatment arm, radiological PFS was 5 months longer versus the monotherapy arm (13.8 vs. 8.2 m). The overall toxicity profile was good, although acute myocardial infarction was reported in 6% of cases [71,72];Abiraterone–Niraparib. This combination is still in the early stages of research. However, promising results have been reported in a phase Ib study that established the doses for future studies.

2.PARP inhibitors combined with Immunotherapy

In preclinical studies, a direct relationship has been found between PARP activity and the regulation of PD-L1 expression, which is maintained when PARP is inhibited. Numerous combinations of PD-1 inhibitors and anti-PD-L1 together with different PARP inhibitors are under investigation [73].

As discussed above, initial studies show an adequate RR (PSA values) when durvalumab and olaparib are combined [58].

Pembrolizumab combined with abiraterone and enzalutamide is also being studied in phase III trial in an unselected population (KEYLYNK-010).

#### 3.2.3. Mechanisms to Sensitize to PARP Inhibitors and Reverse Resistance

In patients without a DDR mutation, the effects of PARP inhibitors are limited; in patient with DDR mutations, the initial effects are attenuated over time due to the emergence of resistance. Several mechanisms of resistance have been described, the most important being the restoration of PARP-independent homologous recombination mechanisms, secondary mutations in PARP, and inactivation of other DNA repair mechanisms. We discuss these below.

One of the first mechanisms of resistance to PARP inhibition discovered was restoration of homologous repair function, independent of PARP activity. There are multiple different pathways leading to the restoration of homologous recombination (HR) function [41], including:

Development of resistance mutations or epigenetic alterations in BRCA 1 and 2, which reverse PARP sensitivity

Alterations in other repair pathways, such as non-homologous recombination or non-homologous end joining, responsible for the repair of DNA double-strand breaks, in a compensatory manner, which stimulates PARP-independent HR. Deficiencies in this repair system can be caused by alterations in various different proteins in this pathway, including such as 53BP1, RAP1 (RIF1) or the ubiquitin ligase RNF8; secondary mutations in RAD51C and RAD51D also lead to the restoration of HR function.

Mutations or epigenetic alterations affecting PARP prevent it from binding to its inhibitors, thus facilitating resistance. Genetic alterations in the PTEN/PI3K/AKT pathway are common in PCa. Mutations in this signaling pathway are responsible for certain cases of resistance to PARP inhibitors.

PTEN, PI3K, and AKT inhibitors have shown good results for the treatment mCRPC with acquired resistance to PARP inhibitors both in monotherapy and as combined treatment with PARP inhibitors [42].

Several strategies are being tested to reverse acquired or initial resistance to this pharmacological group, including:

Restoration of HR appears to be one of the main causes of resistance. Consequently, most research has focused on this area. However, this is a highly complex pathway and there are multiple changes beyond the restoration of BRCA1/2 proteins. Current strategies under active investigation include the addition of HR altering agents such as CDK12, CDK1 and PI3K inhibitors [74].

Another approach is to theoretically-feasible therapeutic combinations such as PI3Kinase inhibitors and combinations involving AKT inhibitors such as ipatasertib (a PTEN inhibitor) [41,75]. There are already promising studies with AKT inhibitors, such as capivasertib in monotherapy. This drug was tested in a phase I basket study of 58 polytreated patients, obtaining a PFS of 6 months. Ipatasertib associated with abiraterone is currently in a phase III study based on promising previous results [76].

Several factors modulate the DDR system, the inhibition of which can lead to BRCAness phenotype [41,77]. The following DDR modulating factors have been described:Hypoxia. This is achieved by associating vascular endothelial growth factor receptor (VEGFR) inhibitors with PARP inhibitors to synergistically inactivate HR activity [78].Other potential targets have been successfully tested in preclinical studies, including ATR, CHK1, WEE1, Aurora kinase, PlK1, and others cell cycle regulators. PK inhibitors of these targets are already available and ready for clinical development [79].

Development of new PARP inhibitors:

Non-NAD-like PARP-1 inhibitors. Classical PARP inhibitors act competitively with the NAD enzyme. This new class of drugs acts by inhibiting PARP binding to histones, but without capturing NAD molecules. In cellular studies of PCa, 5F02 has shown greater activity than classical PARPs against mutated variants such as RA-V7 [80].

Specific PARP-2 inhibitors. Preclinical studies show that PARP-2 and not PARP-1 is responsible for regulating AR transcription by binding to Forkhead Box A1 (FOXA 1). PARP-2 is overexpressed in aggressive forms of PCa. Due to the mechanism of action of PARP-2 inhibitors, these agents do not depend on the DDR system deficiency for effectiveness and fewer adverse events are expected [81].

### 3.3. PSMA-Targeted Therapies

Prostate-specific membrane antigen (PSMA) is a transmembrane glycoprotein expressed in the prostate epithelium. PSMA expression gradually increases from benign epithelium to high-grade intraepithelial neoplasia or prostate carcinoma [82]. Furthermore, PSMA expression appears to be inversely related to androgen levels, with increased activity found in tumor cells that become androgen independent [83]. PSMA is also present in other cancers, specifically in the tumor-associated neovasculature (clear cell renal cancer, transitional cell bladder cancer, testicular-embryonal cancer, neuroendocrine tumors, colon, and breast cancer). However, this binding to the neovasculature does not seem to occur in PCa [83]. Likewise, it appears constitutively in healthy tissues such as lacrimal and salivary glands, epididymis, ovary, normal human prostate epithelium, astrocytes, and Schwann cells in the central nervous system, and within the brush border of the jejunum in the small intestine [84]. In addition, PSMA can function as a receptor that activates signaling to promote cell migration and regulate chromosome stability [85]. In short, PSMA is a highly promising target for the treatment of mCRPC.

The value of PSMA in positron emission tomography-computed tomography (PET-CT) was evaluated prospectively in a study of 314 patients with PCa who developed biochemical relapse after radical treatment. That study confirmed that 68Ga-PSMA PET-CT is superior to choline PET-CT in the restaging of these patients, with a higher diagnostic yield and good safety profile [86].

#### 3.3.1. Radionuclides

Radionuclide therapies consist of radiolabeled small molecules based on glutamate-urea-lysine residues targeting the glutamate carboxypeptidase II pocket of PSMA, or monoclonal antibodies targeting the extracellular domain of the enzyme [86].

The monoclonal antibody 177Lu-PSMA J591 specifically binds to the extracellular domain of PSMA, forming a complex that becomes internalized, releasing the radionuclide into the tumor cells [87]. This antibody can be radioactively labeled and administered in a single dose, and it has shown efficacy in patients with mCRPC (see Table 3).

Phase I studies have determined that the maximum tolerated dose (MTD) of a single dose of 177Lu-PSMA J591 is 70 mCi/m^2^. It can be safely administered in up to three doses of 30 mCi/m^2^ each, which reach common metastatic sites, including bone and soft tissues [88].

In 2013, Tagawa et al. published the results of a study evaluating the efficacy (PSA, RR, and OS) of a single infusion of this antibody. That phase 2 clinical trial included 47 mCRPC polytreated patients in progression. All of the patients had progressed after several lines of hormonotherapy and 55.3% had also received 1 to 4 lines of ChT, including docetaxel. There were two treatment cohorts. One cohort (*n* = 32) was comprised of 15 patients who received 177Lu-PSMA J591 at 65 mCi/m^2^ and 17 patients who received at 70 mCi/m^2^. The other expansion cohort consisted of 15 patients who received 70 mCi/m^2^ to verify RR and to examine biomarkers. PSA levels fell by ≥50% in 10.6% of the patients, by ≥30% in 36.2%, and 59.6% a PSA decline after a single dose of treatment, with a median time to progression of 12 weeks (range, 8 to 47 weeks). Of the 12 patients with measurable disease (25.5% of total), one PR and 8 SD were observed. Grade (G) 4 thrombopenia was observed in 46.8% of cases (no cases of bleeding) and G4 neutropenia in 25.5% (reversible in all cases). Compared to the 65 mCi/m^2^ dose, patients who received the 70 mCi/m^2^ dose (had a further 30% PSA decrease (46.9 vs. 13.3%, *p* = 0.048) and longer OS time (21.8 vs. 11.9 months, *p* = 0.03), but at the cost of greater hematologic toxicity [89].

Given the dose limiting effects of myelosuppression, Tagawa and colleagues conducted a phase 1b/2a dose-escalation study of 177Lu-PSMA J591, under the rationale that dose fractionation may allow safe administration of higher doses. Those authors performed an initial phase 1b dose escalation study (20–45 mCi/m^2^) followed by a phase 2a study involving 49 patients with mCRPC treated with the recommended two doses two weeks apart (40–45 mCi/m^2^). They found that fractionated administration allowed for a higher cumulative radiation dose. PSA levels decrease while OS improved; however, toxicity increased with higher doses (35.3% of patient developed reversible G4 neutropenia and 58.8% thrombocytopenia). Overall, 79.6% of patients had positive PSMA imaging; patients with less intense PSMA imaging tended to have poorer responses [90].

A serious long-term toxicity of this treatment is myelosuppression, although it is predictable and self-limiting, and in most patients does not limit the subsequent administration of other therapies, including ChT [87].

Recently, Niaz et al. hypothesized that additional dose fractionation would allow for a higher cumulative dose, potentially with less toxicity and greater efficacy [91]. Those authors administered 25 mCi/m^2^ every 2 weeks until onset of G2 toxicity in 6 polytreated mCRPC patients (83% were previously treated with abiraterone, 50% with enzalutamide, 50% with docetaxel, 33% with cabazitaxel, 33% with sipuleucel-T, and 17% with investigational agents). Lu-J591 imaging was performed and CTC counts were measured before and after treatment along with a standard follow-up. Patients received three to six doses (cumulative: 75–150 mCi/m^2^). In two (33%) patients, PSA levels decreased by >30% and three patients (50%) showed a decrease in CTC count. A total of two (33%) experienced G 4 neutropenia (no fever), three (50%) had G 4 thrombocytopenia (no bleeding) and two (33%) required platelet transfusions. After hematologic improvement, two patients developed worsening cytopenia during progression of PCa. Bone marrow biopsies revealed the presence of infiltrative tumor replacing normal marrow elements without myelodysplasia. The authors concluded that hyper-fractionation with 177Lu-J591 is feasible, but does not appear to have significant advantages over the two-dose fractionation regimen. Consequently, they recommended single doses of 70 mCi/m^2^ or two fractionated doses of 80–90 mCi/m^2^ (cumulative).

Other ongoing studies are attempting to improve the therapeutic index in combination with other therapies (NCT00859781).

A small molecule targeting PSMA of note is 177Lu PSMA-617 (see Table 4). This drug has shown antitumor activity with less myelosuppression. Since 2015, several retrospective studies have reported promising results with this drug, and a favorable safety profile in patients with mCRPC. The largest study to date was carried out by the German Society of Nuclear Medicine, multicenter, retrospective analysis of 145 polytreated mCRPC patients who received 177Lu PSMA-617 (mean dose, 5.9 GBq). Serial PSA levels (to assess biochemical response) were available for 99 patients, 45% (45 out of 99) of whom had a PSA decrease ≥ 50%. G3-4 hematologic toxicity was observed in 12% (18 out of 145) of the patients [92].

Hofman et al. conducted a prospective phase II study involving 30 patients with mCRPC. Most of the patients (87%) received at least one prior line of ChT (80% docetaxel and 47% cabazitaxel) and 83% had previously received abiraterone or enzalutamide. The mean radioactivity administered was 7.5 GBq per cycle. A PSA decrease of 50% or more was achieved in 57% of patients (17 out of 30). The most common adverse effects were G1 xerostomia (due to PSMA expression in the salivary glands) (87%), G1 and 2 nausea (50%), and G1–G2 nausea (50%). Grade 3–4 toxicity (thrombopenia) occurred in 13% of cases. Objective response of nodal or visceral disease was observed in 14 of 17 patients (82%) with measurable disease and 11 patients (37%) achieved a significant improvement in quality of life (≥10 points on the global health score in the second treatment cycle) [93].

Yadav et al. performed a systematic review and meta-analysis (published in 2019) including 17 studies with 671 patients treated with 177Lu-PSMA-617. That review found that 46% of patients achieved a reduction in PSA values > 50% (and 75% had a decrease in PSA levels post-treatment). The clinical benefit rate overall was 75.5% (37.2% of patients with PR and 38.3% SD). Median OS and PFS were 13.7 and 11 months, respectively, likely due to the fact that the treatment was received in advanced lines. In terms of treatment-related (177Lu-PSMA-617) toxicity, the most common were hematologic toxicity (23% anemia, 14.2% leukopenia, 15% thrombopenia), nephrotoxicity (9.5%), and xerostomia (14.5%). Other toxicities such as fatigue, nausea, and loss of appetite were less common [94].

Recently, Hofman and colleagues published the results of the first analysis of the Thera P study, a randomized phase II trial comparing 177Lu-PSMA-617 (LuPSMA) (6–8 GBq every 6 weeks for up to 6 cycles) to cabazitaxel in 200 men with mCRPC after progression to docetaxel with high PSMA expression on 68Ga-PSMA-11 and 18F- fluorodeoxyglucose (FDG) PET-CT imaging and no FDG-positive/PSMA-negative disease sites vs. cabazitaxel (20 mg/m^2^ every 3 weeks for up to 10 cycles) observing that 177Lu-PSMA-617 achieved a superior rate of PSA reduction ≥50% (primary endpoint) compared to cabazitaxel (66 vs. 37%; *p* < 0.001). In addition, an improvement in PFS (median follow-up 11.3 months) was seen (HR 0.63; 95% CI 0.45–0.88; *p* = 0.007). OS data are still immature. G3–4 adverse events occurred in 32% of men treated with LuPSMA vs. 49% of men treated with cabazitaxel [95].

We are currently awaiting the results of the phase 3 VISION trial. Recruitment has been completed (*n* = 750). That trial is being conducted to evaluate the efficacy of 177Lu-PSMA-617 7, 4 GBq (±10%) every 6 weeks (± one week) for up to 6 cycles, together with best supportive care in patients with mCRPC and 68Ga-PSMA-11 PET-positive, PET-CT in progression who have received ≥ one taxane (maximum two) and a new-generation antiandrogen (enzalutamide and/or abiraterone) in terms of OS and PFS (primary endpoints) vs. best supportive care [96].

Multiple studies investigating the effectiveness and safety of 177Lu-PSMA-617 with dose escalation and in combination with other drugs such as olaparib or pembrolizumab are ongoing (NCT03454750, NCT03042468, NCT03805594, NCT03874884).

#### 3.3.2. Chimeric Antigen Receptor (CAR) T (CAR-T) Cells

Ex vivo modification of autologous T lymphocytes to make them express a chimeric antigen receptor (CAR) can be used to direct the patient’s own lymphocytes to attack the cancer cells. CAR-Ts are genetically modified synthetic molecules in which the effector function of TLs is combined with the ability of antibodies to identify specific tumor-associated antigens. CAR-T cells do not require antigens to be presented by antigen-presenting cells, thus circumventing various immune tolerance mechanisms [97]. CAR-T cells consist of an extracellular domain (involved in antigen identification), a transmembrane domain that contacts the intracellular zone where the immunoreceptor tyrosine-based activation motif (ITAM) is located, which plays a key role in signal transduction aimed at T-cell activation [97]. Currently, in vitro transfection technology is the standard method for transfecting CAR molecules into T lymphocytes.

Despite the advances of CAR-T technology in the treatment of hematological malignancies, in solid tumors they still represent a challenge. In PCa, PSMA has been used as a target for CAR-T cell production.

In 2016, Junghans et al. reported results of the first phase I trial of PSMA-targeted CAR-T cells administered with continuous infusion of low-dose IL-2 [98]. Of the five patients treated in that study, three were successfully engrafted (defined as 20% engraftment of CAR-T cells). No toxicity was reported. In total, two patients achieved partial clinical response, with a reduction of 50 and 70% in PSA values, respectively. The authors suggested that efficacy was limited by low plasma IL-2 depleted by high levels of engrafted activated T-cells, underscoring the impact of “drug–drug” interactions on efficacy.

The immunosuppressive microenvironment, which includes high levels of transforming growth factor beta (TGFβ) found by redirected T cells following tumor infiltration, is also important. The immunosuppressive functions of TGFβ can be overridden in T cells by using a dominant negative TGFβ receptor (TGFβRdn), thereby enhancing antitumor immunity. In vivo models of metastatic PCa have shown that co-expression of TGFβRdn on PSMA-targeted CAR-T cells result in increased T-cell proliferation, increased cytokine secretion, resistance to exhaustion, long-term persistence, and enhanced tumor eradication [99]. Recruitment is now complete for the first phase I clinical trial to evaluate the safety and preliminary efficacy of autologous T-cell modified, CART-PSMA-TGFβRdn cells in men with mCRPC (NCT03089203) [100]. Several other phase I studies are underway to determine the dose of this type of drug (NCT01140373, NCT04227275, NCT04053062, NCT04249947).

#### 3.3.3. Bispecific T-Cell Antibodies

Another approach could be bispecific antibodies, also called “bifunctional” antibodies or bispecific T cell engagers (BITEs) molecules. Bispecific antibodies are antibodies in which two immunoglobulin chains of differing specificity have been fused into a single antibody molecule. This allows the antibody to carry two different antigens in close physical proximity, which in turn can perform a new function, such as immune cells acting on tumor cells or tumor signaling blockade [99].

One example is blinatumomab, a bispecific antibody construct that binds to CD3 on T cells and the cell surface protein CD19, present on precursor B-cells and acute lymphoblastic leukemia (ALL) cells, potentially recruiting cytotoxic T cells to kill ALL cells.

Preclinical models of PCa have shown that AMG 160, an antibody targeting PSMA on PCa cells and CD3 on T cells (leading to T-cell activation, proliferation, and cytokine production) has antitumor activity with specificity for PSMA-positive tumor cells [101]. Pasotuxizumab, another antibody targeting PSMA and CD3, has been evaluated in a phase I dose-escalation study involving 16 patients with mCRPC. The primary objective of that study was to determine safety and MTD. The most common adverse effects ≥ G1 were fever (94%), chills (69%), and fatigue (50%). The most common toxicities ≥ G3 were lymphopenia and infections (both 44%). Dose-dependent clinical activity was observed, with two long-term responders in the dose escalation cohort, with a PSA response of 14 and 19.4 months, respectively. In the latter case, the patient had complete response of soft tissue metastases and partial PR of bone metastases on PSMA-PET-CT imaging, with a >90% reduction in PSA and alkaline phosphatase, and significant and durable improvement in disease-related symptoms [102].

Other phase I studies are currently evaluating these therapies, some in combination with immunotherapy, such as the study evaluating the safety and efficacy of AMG 160 alone and in combination with pembrolizumab (NCT03792841).

Tri-specific T-cell activator models are also being developed. These models, called TriTAC (“Trispecific T-cell Activating Constructs”), have three domains that bind specifically to tumor antigens, human serum albumin, and the CD3-epsilon subunit of the T-cell receptor complex. Austin et al. recently described a novel PSMA-targeted TriTAC (HPN424) in patients with mCRPC [103]. HPN424 is currently being evaluated as monotherapy in a phase I study in patients with mCRPC (NCT03577028).

## 4. Conclusions

Apart from AR blockade—a mainstay in the treatment of castration resistant PCa—and ChT, several additional treatment options are now available for patients with mCRPC, most notably PARP inhibitors and immunotherapy. Understanding the mechanisms of synergy and resistance remains a major challenge, but we have several strategies that are underway where we are seeing that population selection is crucial to improve outcomes. Targeted therapies for PSMA may be another possible treatment option in these patients, particularly radionuclides, which are in more advanced stages of development.

There is a clear need to develop biomarkers to predict response in order to establish the optimal treatment sequence in these patients to extend and improve survival, as well as improving and maintaining quality of life and preventing unnecessary side effects.

## Figures and Tables

**Table 1 biomedicines-09-00392-t001:** Molecular subtypes for prostate cancer (PCa).

Molecular Alteration	Molecular Alteration Subtype	Frequency	Biologic Implication	Proposed Therapy
**Androgen** **Receptor** **(AR)**	Amplification		Resistance to androgen deprivation therapy (ADT)	New antiandrogens (enzalutamide/abiraterone)
	Mutation			Non-NAD-like PARP-1 inhibitors
	Alternative Splicing	4% early stage 20–30% advanced/recurrent disease)		Taxanes
	Changes in the expression of AR Co-regulators			Combine therapies (Antiandrogens + PARPi/Immunotherapy) to modify the immunosuppressive TME
**PI3K-AKT**	Loss of PTEN AKT/PI3K alteration	49%	Resistance to ADT and PARPi	PTEN, AKT and PI3K INHIBITORS (monotherapy/combinations)
**DNA Repair Pathways**	Mutation in the DDR system (homologous repair)		PARPi sensitivity	PARPi
**Others**	Wnt Genetic fusion	23% 18%	Resistance to ADT	Development of new therapeutic targets and combinations

Key: PARPi: PARP inhibitors; TME: tumor microenvironment.

**Table 2 biomedicines-09-00392-t002:** New Therapies for PCa.

Therapy	Type	CSPC/CRPC	FDA Approval	Monotherapy /Combinations
**Parpi**	OLAPARIB	CRPC	YES (in selected population)	Monotherapy Combinations under investigation: + IMMUNOTHERAPY + ANTIANDROGEN+ NEW THERAPIES
	RUCAPARIB	CRPC	YES (in selected population; phase III study is currently underway)	
	NIRAPARIB	CRPC	NO (clinical development program)	
	TALAZOPARIB	CRPC		
**Immunotherapy**	DC VACCINES SIPULEUCEL-T	CRPC	YES	Monotherapy
	ANTI CTLA-4	CRPC	NO (clinical development program)	Combinations under investigation: + PARPi + ANTIANDROGENPembrolizumab in monotherapy (combinations under investigation)
	PD-1/PD-L1 INHIBITORS	CRPC	YES (Pembrolizumab in selected population)	
			NO (other ihibitors in clinical development program)	
**Anti-PSMA** **Therapy**	RADIONUCLIDES	CSPC/CRPC	NO (clinical development program)	Monotherapy
	CAR-T CELLS			
	BISPECIFIC T-CELL ANTIBODIES			

Key: CRPC: castration-resistant prostate cancer; CSPC: castration-sensitive prostate cancer; DC: dendritic cell; PARPi: PARP inhibitors; PSMA: prostate-specific membrane antigen.

**Table 3 biomedicines-09-00392-t003:** Summary of all 177Lu-J591 trials.

Reference	Phase	Patients (*n*)	Dosing Schedule	Treatment (mCi/m^2^)	Biologic Activity and Main Findings
Bander et al. (2005) [88]	Phase I	35	Single	10–75	11.4% PSA declines 46% PSA stabilization 70 mCi/m^2^ was determined to be the single-dose MTD. Multiple doses of 30 mCi/m^2^ are well tolerated.
Tagawa et al. (2013) [89]	Phase 2	47	Single	65–70	59.6% PSA declines 70 mCi/m^2^ resulted in more 30% PSA declines and longer OS.
Tagawa et al. (2019) [90]	Phase 1b/2a dose-escalation	49	Two doses two weeksapart	20–45; 40–45	55.1% PSA declines Fractionated administration allowed higher cumulative radiation dose. The frequency and depth of PSA decrease, OS, and toxicity (dose-limiting myelosuppression) increased with higher doses.
Niaz et al. (2020) [91]	Phase I	6	Every 2 weeks until onset of G2 toxicity	25	33% PSA declines Hyperfractionation is feasible, but does not appear to have significant advantages over the two-dose fractionation regimen

Key: MTD: maximum tolerated dose; PSA: prostate-specific antigen; OS: overall survival.

**Table 4 biomedicines-09-00392-t004:** Summary of 177Lu PSMA-617 studies.

Reference	Study Type	Patients (*n*)	Treatment (GBq)	Biologic Activity
Rahbar et al. (2017) [92]	Retrospective	145	5.9	45% PSA declines
Hofman et al. (2018) [93]	Phase 2	30	7.5	57% PSA declines
Yadav et al. (2019) [94]	Systematic review and meta-analysis	671		75% PSA declines 75.5% clinical benefit rate overall
Hofman et al. (2020) [95]	Randomised phase 2	200 (LuPSMA (*n* = 99) or cabazitaxel (*n* = 101))	6–8	66% PSA declinesImprovement in PSA-PFS

Key: PFS: progression-free survival; PSA: prostate-specific antigen.

## Data Availability

Data sharing not applicable. No new data were created or analyzed in this study.

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
