# Peer review of "Changing the History of Prostate Cancer with New Targeted Therapies"

_biomedicines, 2021, doi:10.3390/biomedicines9040392_

Round 1

Reviewer 1 Report

I enjoyed reading this clinical review article by Hernando-Polo et al., about the recent progress in PCA treatment. Several approaches, other than classical ARPI therapy or in combination with ARPI is elaborated. It mainly discusses recent developments in targeted therapies that include PARP inhibitors and immunotherapy. Approaches for monotherapy and combination treatments are extensively discussed. The authors also provide a comprehensive overview of PSMA targeted therapies that consist of using radionuclides. Recent progress in therapeutic approaches such as CAR-T cells and bispecific T-cell antibodies are also discussed. I found this a very well written article with a wealth of information on clinical management PCA, and provides a broad idea of how the PCA treatment strategies are being evolved in recent times. The article also highlights the importance of discovering new and novel biomarkers to predict therapy response and improve patient survival.

I have a few minor comments.

  1. The authors have to add a logical reason somewhere in the text for the suitability of the current title of this article, for the readers to appreciate the contents.
  2. References are missing in several places; for example, there are no references on pages 1-2, lines 41-52. As well as no references while talking about CRPC phenotype on page 2, lines 54-49.
  3. Please correct "Taxans" to "Taxanes" in Table 1
  4. The sentence “Genetic amplifications of the AR sensitise tumour cells to lower than normal androgen concentrations” on page 2, line 70, is not clear. The sentence can be rewritten as “Genetic amplification of AR will lead to an enhanced expression of AR protein, which in turn make the PCa more sensitive even to a lower concentration of androgen”. These types of sentences are also present in other places of the article, and the authors should reconsider rewriting them to enhance the clarity.
  5. The sentence “AR amplifications have been associated with resistance to hormonal agents” on page 2, line 73, what are the hormonal agents? Please specify a few of them used in anti-hormone therapy.
  6. The sentence “Changes in the expression of AR co-regulators may alter the balance of these co-regulatory proteins, which could help PCa cells to grow” on page 3, line 84-85, what are these co-regulatory proteins? Please provide a few examples or at least a reference.
  7. The sentence “Nevertheless, its development continues, with some studies exploring possible combinations with other medications, most notable highly sipuleucel- T associated with radium 223” on page 5, lines 168-169, this sentence is not clear.
  8. The sentence “DNA vaccines using a bacterial plasmid with an encrypted tumour antigen” on page 5, line 184, is there a reference for this study? or is only preliminary data available?
  9. I found the section on “3.3.1. Radionuclides” on page 12 onwards a bit dense in text. Is it possible to illustrate the major developments in this area of research in the form of a table, for a faster perception of the content? This is a suggestion, otherwise, it looks fine.
  10. Word “highly” is repeated on page 5, line 192.
  11. Correct “appearsfor” to “appears for” on page 5, line 192.
  12. Add reference for “Pritchard et al” on page 7, line 289.

Author Response

Dear editor

First of all we should like to thank the reviewers for their comments to improve the work.

We have modified the manuscript considering all the comments using track changes.

REVIEWER 1

  1. The authors have to add a logical reason somewhere in the text for the suitability of the current title of this article, for the readers to appreciate the contents.

Title has been modified.

  1. References are missing in several places; for example, there are no references on pages 1-2, lines 41-52. As well as no references while talking about CRPC phenotype on page 2, lines 54-49.

The following references have been added (highlighted in the manuscript):

  • [3]- Wang G, Zhao D, Spring DJ, DePinho RA. Genetics and biology of prostate cancer. Genes Dev. 2018 Sep 1;32(17-18):1105-1140. (Pag 1)
  • [4]- Hong JH, Kim IY. Nonmetastatic castration-resistant prostate cancer. Korean J Urol. 2014 Mar;55(3):153-60 (Pag 2)

  1. Please correct "Taxans" to "Taxanes" in Table 1

It has been corrected.

  1. The sentence “Genetic amplifications of the AR sensitise tumour cells to lower than normal androgen concentrations” on page 2, line 70, is not clear. The sentence can be rewritten as “Genetic amplification of AR will lead to an enhanced expression of AR protein, which in turn make the PCa more sensitive even to a lower concentration of androgen”. Thank you for the suggested sentence. It has been modified.

These types of sentences are also present in other places of the article, and the authors should reconsider rewriting them to enhance the clarity.

  1. The sentence “AR amplifications have been associated with resistance to hormonal agents” on page 2, line 73, what are the hormonal agents? Please specify a few of them used in anti-hormone therapy.

Some examples have been added.

  1. The sentence “Changes in the expression of AR co-regulators may alter the balance of these co-regulatory proteins, which could help PCa cells to grow” on page 3, line 84-85, what are these co-regulatory proteins? Please provide a few examples or at least a reference.

Some examples have been added.

  1. The sentence “Nevertheless, its development continues, with some studies exploring possible combinations with other medications, most notable highly sipuleucel- T associated with radium 223” on page 5, lines 168-169, this sentence is not clear.

The sentence has been rewritten.

  1. The sentence “DNA vaccines using a bacterial plasmid with an encrypted tumour antigen” on page 5, line 184, is there a reference for this study? or is only preliminary data available?

The following reference has been added (highlighted in the manuscript):

Reference [28]: Zahm, C. D., Colluru, V. T., & McNeel, D. G. (2017). DNA vaccines for prostate cancer. Pharmacology & therapeutics, 174, 27–42. https://doi.org/10.1016/j.pharmthera.2017.02.016.

  1. I found the section on “3.3.1. Radionuclides” on page 12 onwards a bit dense in text. Is it possible to illustrate the major developments in this area of research in the form of a table, for a faster perception of the content? This is a suggestion, otherwise, it looks fine.

Thanks for your suggestion. Two tables have been included  in that section (“3.3.1. Radionuclides”): Table 3 and Table 4.

  1. Word “highly” is repeated on page 5, line 192.

It has been removed.

  1. Correct “appearsfor” to “appears for” on page 5, line 192.

It has been corrected adding a space between the words.

  1. Add reference for “Pritchard et al” on page 7, line 289.

It has been added: Reference [45] (highlighted in the manuscript): Pritchard CC, et al. Inherited DNA-Repair Gene Mutations in Men with Metastatic Prostate Cancer. N Engl J Med. 2016 Aug 4;375(5):443-53. doi: 10.1056/NEJMoa1603144. Epub 2016 Jul 6. PMID: 27433846; PMCID: PMC4986616.

We look forward to hearing from you in due time regarding our submission and to respond to any further questions and comments you may have. Sincerely,

Susana Hernando y Diana Moreno.

Reviewer 2 Report

In this review article, Hernando Polo et al.  review how the therapeutic landscape of metastatic castration-resistant prostate cancer (mCRPC) is changing due to the emergence of new targeted therapies for the treatment of different molecular 20 subtypes. They also summerize different novel strategies to improve outcomes in patients with prostate cancer.

This is a very interesting article summarizing plenty of interesting data on research and trials of the new targeted therapies.

The manuscript title is not very well describing the content. The title could be modified e.g. by addition of “with new targeted therapies” or alike, to better reflect the topic at hand.

The manuscript contains paragraphs of only single or double sentences at disturbingly many occasions. Even though in list-like points this may be justified, there are several points where these could easily be fused to the surrounding paragraphs, or by slight modification of the order of things presented, to fuse several “orphan topics” to same paragraphs.

Table 1 should be made more informative by adding the frequencies of the alteration in the main categories (despite that many are listed in the additional text).

In association to Table 1, it is mentioned that “gene fusion is another aberration present in CRPC”. It would be worthy of mentioning that TMPRSS2-ERG fusion is the single most frequent genetic alteration in prostate cancer, and the gene fusions in general are highly prevalent in PCa. Although this aberration is not directly linked to PCa drug treatment stratification or formation of resistance, it is a significant part of subtyping of tumors in the literature and has clinical significance in advanced PCa.

Author Response

Dear editor

First of all we should like to thank the reviewers for their comments to improve the work.

We have modified the manuscript considering all the comments using track changes.

REVIEWER 2

  • The manuscript title is not very well describing the content. The title could be modified e.g. by addition of “with new targeted therapies” or alike, to better reflect the topic at hand.

Thank you for the suggested title. It has been modified.

  • The manuscript contains paragraphs of only single or double sentences at disturbingly many occasions. Even though in list-like points this may be justified, there are several points where these could easily be fused to the surrounding paragraphs, or by slight modification of the order of things presented, to fuse several “orphan topics” to same paragraphs.

This finding has been corrected.

  • Table 1 should be made more informative by adding the frequencies of the alteration in the main categories (despite that many are listed in the additional text).

A new column has been added including this information.

  • In association to Table 1, it is mentioned that “gene fusion is another aberration present in CRPC”. It would be worthy of mentioning that TMPRSS2-ERG fusion is the single most frequent genetic alteration in prostate cancer, and the gene fusions in general are highly prevalent in PCa. Although this aberration is not directly linked to PCa drug treatment stratification or formation of resistance, it is a significant part of subtyping of tumors in the literature and has clinical significance in advanced PCa.

Sentence has been modified adding this clarification.

We look forward to hearing from you in due time regarding our submission and to respond to any further questions and comments you may have. Sincerely,

Susana Hernando y Diana Moreno.
